

# Entanglement in a free fermion chain under continuous monitoring

Xiangyu Cao[1⋆], AntoineTilloy[2] and Andrea De Luca[3]

**1** Department of Physics, University of California, Berkeley, Berkeley CA 94720, USA
**2** Max-Planck-Institut für Quantenoptik,
Hans-Kopfermann-Straße 1, 85748 Garching, Germany
**3** The Rudolf Peierls Centre for Theoretical Physics,
Oxford University, Oxford, OX1 3NP, United Kingdom

⋆ xiangyu.cao@berkeley.edu

## Abstract

We study the entanglement entropy of the quantum trajectories of a free fermion chain under continuous monitoring of local occupation numbers. We propose a simple theory for entanglement entropy evolution from disentangled and highly excited initial states. It is based on generalized hydrodynamics and the quasi-particle pair approach to entanglement in integrable systems. We test several quantitative predictions of the theory against extensive numerics and find good agreement. In particular, the volume law entanglement is destroyed by the presence of arbitrarily weak measurement.

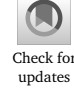
# 1 Introduction

The dynamics of entanglement in many-body systems is a topic under intensive study, as entanglement is a fundamental notion of quantum physics, deeply tied to basic issues such as thermalization of closed quantum systems [1–4], holography [5–7], quantum chaos and information scrambling [8, 9]. Understanding entanglement is also crucial for designing effective quantum simulators and determining the limits thereof [10–12]. Thanks to recent experimental progress, these fundamental issues can now be addressed in a laboratory [13–15].

The subtle rôle played by measurement is another central feature of quantum mechanics, and has stimulated debates over interpretations for almost a century [16–18]. Stepping aside from foundational considerations, it is important to understand the interplay between measurement dynamics and unitary system dynamics in practice. Such a study is made possible with the help of weak and continuous measurements, which enable a non-destructive probing of quantum systems. Continuous measurements can now be realized experimentally, notably in superconducting circuits [19–21]. The resulting stochastic quantum trajectories (QT) have been well understood theoretically for finite dimensional Hilbert spaces, especially in the Zeno limit of strong measurement, where jumps [22] and spikes [23, 24] between measurement pointer states emerge. (See also [25] for a review on the related quantum-jump approach and [26] for recent developments.)

A natural question arises where the two important concepts meet: what is the dynamics of entanglement in many-body systems under continuous measurement? Recently, there has been a spike of interest [27–30] in understanding the scaling law of the entanglement entropy in such a setting where an extensive amount of measurement, which tends to destroy all entanglement, competes with a thermalizing unitary dynamics (which tends to generate a volume law entanglement). An exciting suggestion [27, 30] was made of an entanglement transition in the thermodynamic limit, as one tunes the measurement strength across a nonzero threshold below which the volume law would still survive. To date, the existence of such a volume law phase is not completely settled [29]. More generally, it is fair to say that the entanglement structure of many-body QT's is far from being sufficiently understood (results do exist for few-body systems [31]). The rôle played by different types of unitary dynamics and measurement awaits to be elucidated as well.

Pure state stochastic dynamics – *unravelling* – can also be used to numerically simulate the dynamics of an open quantum system [32]. In this context, the stochastic trajectories need not have any connection with a physical measurement setup, and different unravelings can reduce to the same Lindblad equation upon averaging. One may then pick the easiest to simulate. Since entanglement represents the main bottleneck for simulability, understanding the entanglement growth for different unravelings can help assess which is the most appropriate.

In this paper, we address these issues in the specific case where the many-body system is non-interacting (free), exemplified by a simple model of many-body QT: a one-dimensional chain of free fermions whose local occupation numbers are continuously measured. We note that numerous variants of this model have been studied [33–37] using an open quantum system approach [38–40], which focuses on the Lindblad equation, and its large deviation theory extension [26, 35, 37]. These methods have been used to compute observables of the averaged density of state ("mean state"), for instance, the integrated current statistics in an non-equilibrium stationary state. However, these results do not reveal the entanglement dynamics of the individual QT's, also known as the "conditional state". (The entanglement of the mean state could certainly be derived from these earlier results, but is a less interesting and in any case different quantity.) At the QT level, the model was addressed before us only in the strong measurement limit [41]; therefore, beyond this limit, the behavior of non-linear observables of the density of state, including the notable example of entanglement entropy, has not been

understood. Here, our primary goal is the study of entanglement dynamics for weak to intermediate measurement, in the stationary regime and during post-quench transients.

To make precise predictions about entanglement entropy, we adopt the approach of generalized hydrodynamics (GHD) and the associated quasiparticle pair picture [42–44]. GHD has been successfully developed as a large-scale description of the non-equilibrium dynamics of Bethe-integrable systems, both free and interacting [45–48]. From a large variety of initial conditions (with the notable exception of zero-entropy stationary states [49]), the entanglement dynamics is entirely produced by spatially separated quasiparticle pairs [44, 50–52]. They travel ballistically and independently [1]. For our model, we derive an exact GHD description of the averaged density of state, which is equivalent to the known exact solution [33, 35]. To go beyond that and study the entanglement dynamics of individual QT's, we propose the following Ansatz: continuous measurement collapses randomly (at a rate given by the measurement strength) quasiparticle pairs and creates a new pair in place, as illustrated in Fig. 1. This "collapsed quasiparticle pair Ansatz" is motivated by GHD, and allows us to make *quantitative* predictions on entanglement dynamics. We compare them to extensive numerical simulations, which are possible thanks to the remarkable fact that the non-linear QT's preserve Gaussianity. The Ansatz turns out to be surprisingly effective given its simplicity. Its predictions are quantitatively accurate in many interesting situations, and captures the main features of continuously measured non-interacting models. Namely, the volume law entanglement is destroyed by arbitrarily weak continuous measurement (performed on all degrees of freedom). Indeed, since quasiparticle pairs have a typical lifetime $1/\gamma$, where $\gamma$ is the measurement rate, entanglement cannot be built beyond the length scale $v_{max}/\gamma$ where $v_{max}$ is the maximal quasiparticle velocity. Such a picture is general for free fermions systems, even in higher dimensions. As we discuss further in Section 6, a similar behavior is expected in integrable systems. Such an absence of volume law in the presence of even a weak measurement is in sharp contrast with the (putative) entanglement transition in thermalizing systems discussed above. So, although both a free Hamiltonian and chaotic one can generate a volume law entanglement, the nature of the latter is very different, and can be revealed by whether it is stable under weak measurement.

The rest of the paper is organized as follows. In Section 2, we define the model and discuss briefly the problem under consideration. In particular, we explain the relation to Lindblad equation, as well as the distinction between "mean state" and "conditional state". Section 3 describes the solvability of the model and the numerical simulation method. Section 4 derives the generalized hydrodynamics equation, and illustrate it with two examples (heating and transport). Section 5 contains the main contribution of this work: it describes in detail the collapsed quasiparticle pair Ansatz for entanglement entropy and test its validity in several situations. We close with some concluding discussions and preliminary results of further work in Section 6.

## 2 Model

We start with a model of free fermions on a one-dimensional lattice with Hamiltonian:

$$H = \lambda \sum_{i=1}^{L} \left[ a_{i+1}^{\dagger} a_i + a_i^{\dagger} a_{i+1} \right],\tag{1}$$

where $\lambda$ is the coupling constant $a_i^{\dagger}, a_i$ are the fermionic creation and annihilation operators on site $i$. We assume periodic boundary conditions ($i + L \equiv i$) when considering homogeneous initial condition, while open boundary condition is considered for two-reservoir initial

---

[1]This is true in the non-interacting case. An integrable interaction would generate a nontrivial phase shift.

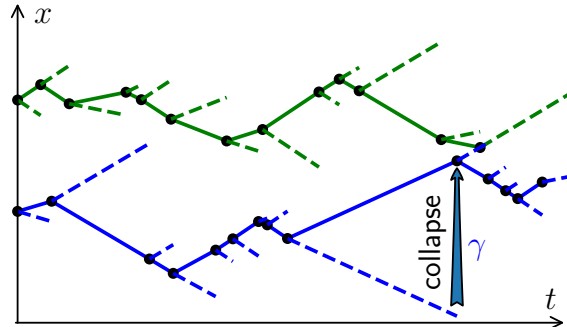

Figure 1: Illustration of the collapsed quasiparticle pairs Ansatz describing the entanglement dynamics of a free fermion chain under continuous measurement of local occupation numbers with rage $\gamma$. The initial state ($t = 0$) is a product state, described by pairs of localized quasiparticle pairs with opposite velocities (only two pairs are depicted). A quasiparticle travels ballistically and independently, until it is collapsed by the measurement (this happens at rate $\gamma$ for each quasiparticle, independently of others). When that happens, a new pair is created at the position of its partner, with velocity $v = \pm v(k)$ where $k$ is a randomly chosen momentum (uniform in $[-\pi, \pi)$) and $v(k)$ is its group velocity. See Sections 4 and 5 for further detail.

conditions. We want to add to this unitary evolution the dynamics induced by the continuous measurement of the local fermion numbers $n_i = a_i^\dagger a_i$. For a general observable (self-adjoint operator) $\mathcal{O}$, the associated monitored dynamics of the quantum state is described by the following stochastic Schrödinger equation (SSE) [53–55]:

$$d|\psi_t\rangle = -\mathbf{i}H dt|\psi_t\rangle + \left(\sqrt{\gamma}\,[\mathcal{O} - \langle\mathcal{O}\rangle_t]\,dW_t - \frac{\gamma}{2}[\mathcal{O} - \langle\mathcal{O}\rangle_t]^2\,dt\right)|\psi_t\rangle, \qquad (2)$$

where $\langle\cdot\rangle_t = \langle\psi_t|\cdot|\psi_t\rangle$, $W_t$ is a Wiener process (Brownian motion), and $\gamma$ denotes the measurement strength or rate. The multiplicative noise is understood in the Itô convention [56]. Note that the contribution from the continuous measurement process is described by the second term of (2), and can be realized by homodyne detection in quantum optics [57] (see also Ref. [58] for a proposal in the cold atom context), and more generally from infinitely weak and frequent interactions with ancillas which are then projectively measured [59–61]. In the present many-body setting, we will study the simultaneous monitoring of the occupation number of all sites, which is possible because the operators $n_i$'s commute with each other. The corresponding SSE is the following:

$$d|\psi_t\rangle = -\mathbf{i}H dt|\psi_t\rangle + \sum_{i=1}^{L}\left(\sqrt{\gamma}[n_i - \langle n_i\rangle_t]\,dW_t^i - \frac{\gamma}{2}[n_i - \langle n_i\rangle_t]^2\,dt\right)|\psi_t\rangle, \qquad (3)$$

where $W_t^i, i = 1,\ldots,L$ are independent Wiener processes.

We observe that the total particle number $\sum_i n_i$ is conserved by the SSE (3), in the following sense: if the initial state $|\psi_{t=0}\rangle$ has a fixed number of total particle number,

$$\sum_i n_i|\psi_{t=0}\rangle = N|\psi_{t=0}\rangle, \qquad (4)$$

so does $|\psi_t\rangle$ for any $t > 0$ and for any realization of $\{W_t^i\}$. Therefore, in the following, we will always assume (4) with a finite filling rate, $N/L = $ const.

## 2.1 Lindblad equation, mean state and conditional state

Before going on, it is helpful to place our model in the more general context of open quantum systems, and discuss the distinction between two important notions: "mean state" and "conditional state".

The *mean state* (also called the a priori state) is defined as the density matrix averaged over measurement outcomes:

$$\overline{\rho_t} := \overline{|\psi_t\rangle\langle\psi_t|}, \tag{5}$$

where $\overline{[\dots]}$ denotes the average over measurement outcomes. It is well known that the SSE (2) implies (by Itô calculus) the following Lindblad master equation for $\overline{\rho_t}$:

$$\partial_t\overline{\rho_t} = \mathcal{L}\overline{\rho_t} = -\mathbf{i}[H,\overline{\rho_t}] - \gamma\sum_j\frac{1}{2}[n_j,[n_j,\overline{\rho_t}]]. \tag{6}$$

The generator of time evolution $\mathcal{L}$ is a linear super-operator. The dynamics of the mean state of our model (as well as its variants) has been extensively studied [33,35,36]. The continuous measurement SSE (3) is known to be an *unravelling* of the Lindblad equation (6). There exist other unravellings, i.e., SSE's that lead to the same Lindblad equation for the mean state, as we discuss further in Section 6; in this paper, we focus on the particular unravelling (3).

The *conditional state* (or a posteriori state) is the QT $|\psi_t\rangle$ itself, or the density matrix $\rho_t = |\psi_t\rangle\langle\psi_t|$ *without* averaging over measurement. To better appreciate the distinction, let $\mathcal{O}[\rho_t]$ denote a functional of $\rho_t$. It follows that

$$\overline{\mathcal{O}[\rho_t]} = \mathcal{O}[\overline{\rho_t}] \text{ if } \mathcal{O} \text{ is linear in } \rho_t. \tag{7}$$

This is the case for the quantum expectation value of any operator, e.g., the occupation number $\mathcal{O}[\rho_t] := \text{Tr}[\rho_t n_j]$; then the average over QT

$$\overline{\langle n_j\rangle}_t = \overline{\mathcal{O}[\rho_t]} = \text{Tr}[\overline{\rho_t}n_j]$$

can be calculated from the sole knowledge of the mean state. However, for *non-linear* functionals, $\overline{\mathcal{O}[\rho_t]} \neq \mathcal{O}[\overline{\rho_t}]$ in general: its average over QT's cannot be obtained from the mean state. As a simple example, consider the purity $\mathcal{O}[\rho_t] := \text{Tr}[\rho_t^2]$. Since the conditional state $\rho_t = |\psi_t\rangle\langle\psi_t|$ remains pure (for any measurement outcome), $\overline{\mathcal{O}[\rho_t]} = \overline{\text{Tr}[\rho_t^2]} = 1$. Yet, the mean state $\overline{\rho_t}$ will generically become mixed for $t > 0$ [see (9) below], so $\mathcal{O}[\overline{\rho_t}] = \text{Tr}[\overline{\rho_t}^2] < 1$. In general, the evolution of nonlinear functionals of the conditional state is significantly more complex than that of the mean state; for recent related works, see Refs. [62,63] [2].

## 2.2 Entanglement entropy: a first look

The main object of this work is a well-known nonlinear functional of the conditional state $\psi_t$, the entanglement entropy (EE) between an interval $[x_1, x_2]$ and the rest of the system. We recall that it is defined as the von Neumann entropy of the reduced density matrix $\rho_{[x_1,x_2]}$

$$S_{[x_1,x_2]}(t) := -\text{Tr}\left[\rho_{[x_1,x_2]}\log_2\rho_{[x_1,x_2]}\right], \text{ where } \rho_{[x_1,x_2]} := \text{Tr}_{[1,L]\backslash[x_1,x_2]}[|\psi_t\rangle\langle\psi_t|], \tag{8}$$

where $\text{Tr}_{[1,L]\backslash[x_1,x_2]}$ denotes a partial trace on the complement of the interval. (Note that we measure entropy in base-2 logarithms throughout this paper.) We will be interested in $\overline{S_{[x_1,x_2]}}(t)$, which is the EE of the conditional state, averaged over all quantum trajectories (measurement outcomes).

---

[2]We thank the Anonymous Referee for pointing out an error in the previous version of the manuscript. The current revision is inspired by his/her helpful remarks.

The EE (8) is a highly non-linear functional of $\rho_t$, and it is impossible to infer its behavior from the known results on $\overline{\rho_t}$. To see this more concretely, let us consider the stationary state, reached in the long time limit $t \to \infty$. It is known [33, 35, 36] that for any $\gamma > 0$, the $\overline{\rho_t}$ tends to the fully mixed (infinite-temperature) state with fixed total particle number:

$$\overline{\rho_t}|_{t\to\infty} \propto \sum_{j_1 < \cdots < j_N} a_{j_1}^\dagger a_{j_2}^\dagger \ldots a_{j_N}^\dagger |\Omega\rangle\langle\Omega| a_{j_N} \ldots a_{j_2} a_{j_1} , \qquad (9)$$

where $|\Omega\rangle$ is the vacuum. Therefore the mean state entanglement in the long time limit is independent of $\gamma$ [3]. In contrast, the entanglement of the conditional state in the long time limit depends non-trivially on $\gamma$, and so does the average $\overline{S_{[x_1,x_2]}(t)}$, which will be studied quantitatively in Section 5. Nevertheless, we can already get some first ideas by considering the two extreme cases:

- When $\gamma/\lambda = 0$, we have a free fermion chain, whose entanglement dynamics is well studied. For instance, starting from the Néel product state

$$\left[\prod_{j=1}^{L/2} a_{2j}^\dagger\right] |\Omega\rangle . \qquad (10)$$

The EE of any interval of length $\ell$ grows linearly in time $dS/dt = \frac{4}{\pi}\lambda$ until saturating a volume $S \sim \ell$ at time $t \sim \ell/\lambda$ [65]. In the long time limit, the system reaches a generalized Gibbs equilibrium uniform (GGE) [66]. We refer to Section 5.1 below for further discussion on entanglement, and to Ref. [67] for a review on quench dynamics.

- When $\gamma/\lambda = +\infty$ (i.e., in absence of unitary evolution), the QT converges to a random choice of the $L!/(N!(L-N)!)$ pointer states $a_{j_1}^\dagger \ldots a_{j_N}^\dagger |\Omega\rangle$, $j_1 < \cdots < j_N$, destroying all entanglement. Adding a small hopping term $0 < \lambda \ll \gamma$ will generate classical stochastic jumps between pointer states, leading to a symmetric simple exclusion process (SSEP) [41]. Note that these behaviors are not specific to the free Hamiltonian chosen, and will remain essentially unchanged by adding interaction terms which preserve integrability ($\sum_j n_j n_{j+1}$) or break it ($\sum_j n_j n_{j+2}$).

Given the distinct behaviors in the two limits, one naturally wonders what happens in the intermediate parameter regimes $\gamma \sim \lambda$. In particular, does the entanglement at the long time limit obey a volume law or an area law? These are the main goals of our study.

Since the behavior of the system depends only on the ratio $\gamma/\lambda$ (and the dimensionless time $\lambda t$) we shall fix $\lambda = 1/2$ in what follows.

## 3 Numerical method

A remarkable property of Eq. (3) is that it preserves the Gaussianity of QT. Indeed, since $n_i^2 = n_i$ for fermions, the evolution generator is quadratic. Note that this property would be lost averaging over the measurement results, although the resulting Lindblad equation still enjoys

---

[3]There are a few different ways of defining the entanglement entropy for mixed states, see *e.g.* [64]. They all reduce to the von Neumann entropy (vNE, 8) when applied to a pure state. For example, the *entanglement of formation* of a mixed state is defined as the least expected entanglement entropy of any ensemble of pure states realizing it [64]. By this definition, the fully mixed state (9) has zero entanglement, since it is explicitly realized by an ensemble of product states. Note also that for mixed states, the vNE (of reduced density matrix) is not a measure of entanglement: the fully mixed state (9) has maximal vNE.

exact solvability but in a more complicated sense [33, 36]. Now, a Gaussian state is fully characterized by its two point function

$$D_{ij}(t) = \langle a_i^\dagger a_j \rangle_t. \tag{11}$$

It satisfies a closed equation, which can be derived from Eq. (3). For this, we assume the particle number is fixed in the Gaussian state: this is done by considering particle number conserving Gaussian initial states, since the Hamiltonian (1) preserves this property for later times. Then, using Itô's lemma and Wick's theorem for Gaussian states ($\langle \psi_t | a_i^\dagger a_j a_k^\dagger a_l | \psi_t \rangle = D_{ij}(t) D_{kl}(t) + D_{il}(t)(\delta_{kj} - D_{kj}(t))$ [68]), after some routine algebra, we obtain the following stochastic differential equation:

$$dD = -\mathbf{i}[h, D]\,dt - \gamma(D - D^{\mathrm{diag}})\,dt + \sqrt{\gamma}\,[D\,dW + dW\,D - 2D\,dW\,D]. \tag{12}$$

Here, $[h]_{kl} := \frac{1}{2}(\delta_{k,l+1} + \delta_{k,l-1})$ is the hopping matrix corresponding to the Hamiltonian (1), and $[D^{\mathrm{diag}}]_{kl} := \delta_{k,l}D_{kl}$ is the diagonal part of $D$, and $[W]_{kl} := \delta_{k,l}W^l$. Note that Eq. (12) extends readily to arbitrary quadratic Hamiltonians, by changing the hopping matrix $h$.

In principle, Eq. (12) makes it possible to perform extensive ($L \geq 10^2$) and exact numerical simulations in any spatial dimension. Since we shall focus on pure Gaussian states, we use a more efficient and reliable numerical scheme, which represents the state by an $L \times N$ matrix $U$:

$$|\psi_t\rangle = |U\rangle := \prod_{k=1}^{N}\left[\sum_{j=1}^{L} U_{jk} a_j^\dagger\right]|\Omega\rangle. \tag{13}$$

We require that $U$ be an isometry: $U^\dagger U = \mathbf{1}_{N \times N}$. Then the correlation matrix is given by $D = U U^\dagger$. (The filling fraction $N/L$ is fixed when considering the $L \to \infty$ limit.) To simulate the dynamics, we Trotterize Eq. (3) by alternating its unitary and measurement terms, with time step $\delta t$:

$$|\psi_{t+\delta t}\rangle \approx C e^{\sum_j\left[\delta W_t^j + (2\langle n_j\rangle_t - 1)\gamma\delta t\right]n_j} e^{-\mathbf{i}H\delta t}|\psi_t\rangle, \tag{14}$$

then where $\delta W_t^j$ are independent, each with zero mean and $\gamma\delta t$ variance, and $C$ is a normalization constant. It follows that:

$$|\psi_{t+\delta t}\rangle \approx C|V_{t+\delta t}\rangle, \quad \text{where } V_{t+\delta t} = M e^{-\mathbf{i}h\delta t} U \text{ and } M_{jk} = \delta_{jk} e^{\delta W_t^j + (2\langle n_j\rangle_t - 1)\gamma\delta t}. \tag{15}$$

Above, $|V_{t+\delta t}\rangle$ is defined by the matrix $V$ in the same way as (13); $h$ is the hopping matrix (12) and $\langle n_j\rangle_t = \sum_k U_{jk}U_{jk}^*$ can be readily computed from $U$. To finish the integration over the time step $[t, t+\delta t]$, we perform a QR decomposition $V_{t+\delta t} = QR$, and set $U_{t+\delta t} := Q$, restoring the isometry property. The main advantage of this method [compared to solving (12) directly] is that the purity of the Gaussian state is conserved by construction. Therefore, the time step $\delta t$ does not need to be very small to avoid non-physical errors. In practice, $\delta t = 0.05$ is sufficiently accurate for our purposes. (We checked that the numerical data below are not affected if $\delta t$ is doubled or halved.)

Finally we recall that the entanglement entropy can be readily computed in terms of the two-point function matrix $D$ defined in (11). In particular, to calculate the EE between the sub-system $[x_1, x_2]$ and the rest of the system, one simply needs to diagonalize the sub-matrix $\left[D_{ij}(t)\right]_{i,j=x_1}^{x_2}$. Denoting its eigenvalues by $\lambda_1, \ldots, \lambda_\ell$, where $\ell = x_2 - x_1 + 1$, the entanglement entropy (8) is given by [43, 65, 69]

$$S(t) = S_{[x_1,x_2]}(t) = -\sum_{j=1}^{\ell}\left[\lambda_j \log_2(\lambda_j) + (1 - \lambda_j)\log_2(1 - \lambda_j)\right]. \tag{16}$$

This formula allows efficient numerical calculation of EE of a QT obtained from direct simulation. Yet, it does not allow to understand the time evolution of EE in large systems, even qualitatively. To do this we shall adopt the generalized hydrodynamics (GHD) approach.

# 4 Generalized Hydrodynamics

In this section, we derive a generalized hydrodynamic description (GHD) of the model. This is an exact description of the mean state, which is, mathematically speaking, insufficient for the computation of conditional state entanglement (see Section 2). However, the GHD provides a physical description of the model in terms of quasiparticles, which will be crucial for understanding the entanglement dynamics (see Section 5 below).

The GHD is a continuum equation that describes the evolution of the quasiparticle momentum distribution. For free-fermion systems, the latter is defined as the Wigner distribution

$$n(x, k, t) := \sum_s e^{iks} D_{x-s/2, x+s/2}(t),\tag{17}$$

where $D$ is the matrix of correlations defined in Eq. (11). Strictly speaking, $n(x, k, t)$ is only defined when $2x$ is an integer and the sum over $s$ runs over values for which $x \pm s/2$ are integers; in a finite system, the number of independent momenta $k$ is also finite. As is common in hydrodynamic approaches, we consider profiles $n(x, k, t)$ which are slowly varying functions of $x, k$ so that they can be treated as continuous variables, $x \in \mathbf{R}$ and $k \in [-\pi, \pi]$. Furthermore, we first consider the average of Eq. (12), i.e., we discard the $dW$ term. Combining it with the definition of $n(x, k, t)$, we can obtain the following GHD equation:

$$\partial_t n(x, k, t) = -v(k)\partial_x n(x, k, t) - \gamma\left(n(x, k, t) - \int_{-\pi}^{\pi} \frac{dk}{2\pi} n(x, k, t)\right),\tag{18}$$

where

$$v(k) = -i\lambda(e^{ik} - e^{-ik}) = \sin(k)\tag{19}$$

is the quasiparticle velocity (recall $\lambda = \frac{1}{2}$). The explicit form of (19) depends on the hopping matrix $h$ in (12), of which (18) is independent. We note that in deriving (18) [from (12) and (17)] consists of replacing the lattice derivative by the continuum one: $n(x \pm \frac{1}{2}, k, t) - n(x, k, t) \approx \frac{1}{2}\partial_x n(x, k, t)$. We refer to Ref. [70] for further discussion on the accuracy of this approximation and higher corrections.

The GHD equation (18) is a linear equation, and admits the following probabilistic interpretation: $n(x, k, t)$ is the joint distribution of the position and momentum of non-interacting classical quasiparticles. They propagate with velocity $v(k)$ [first term of (18), RHS] but have a probability $\gamma dt$ in every interval $dt$ of picking a new random momentum $k' \in [-\pi, \pi]$ with uniform distribution [second term of (18), RHS]. The latter is the effect of the continuous measurement, and breaks the conservation (under free fermion evolution) of quasiparticle density at fixed momentum $N(k, t) = \int n(x, k, t)dx$.

In the remainder of this section, we consider two applications that do not concern entanglement. We note that both examples can be treated with existing methods [33, 35]; the goal here is to familiarize ourselves with the GHD point of view, which is useful for the study of entanglement in Section 5 below.

The first is the measurement induced heating from a random eigenstate $|\psi_0\rangle$ of $H$ with inverse temperature $\beta$ and zero chemical potential. It corresponds to a homogeneous initial profile

$$n(x, k, t = 0) = \frac{1}{2(1 + e^{\beta \epsilon(k)})} \text{ where } \epsilon(k) = \cos(k)\tag{20}$$

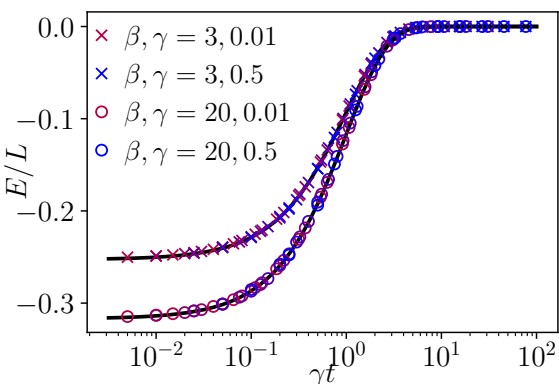

Figure 2: Heating by continuous measurement from a lower energy eigenstate at inverse temperature $\beta = 3$ and $\beta = 20$. It is chosen by randomly filling the orbital with momentum $k$ with probability given by the Fermi factor $\frac{1}{2}(1 + e^{\beta \epsilon(k)})^{-1}$. The average energy density $E/L = \langle \psi_t | H | \psi_t \rangle / L$ is plotted for measurement rates are $\gamma = 0.01, 0.02, 0, 05, 0.1, 0.2, 0.5$ (markers, from red to blue), as a function of rescaled time $\gamma t$. The predictions from GHD (21) are plotted in solid curves.

is the quasiparticle energy. Then (18) implies that $n(x, k, t) = \frac{1}{2} + e^{-\gamma t}(n(x, k, 0) - \frac{1}{2})$, i.e., the Fermi distribution tends exponentially towards the infinite-temperature stationary state of the model, with the characteristic time $1/\gamma$. The energy density behaves thus similarly:

$$E(t)/L = \frac{1}{L} \int_0^L dx \int_{-\pi}^{\pi} \frac{dk}{2\pi} n(x, k, t) \epsilon(k) = e^{-\gamma t} \int_{-\pi}^{\pi} \frac{dk}{2\pi} \frac{1}{2} \frac{\epsilon(k)}{1 + e^{\beta \epsilon(k)}} . \qquad (21)$$

Fig. 2 compares this prediction with the average expectation value $\overline{\langle \psi_t | H | \psi_t \rangle}/L$ from numerical simulation, and obtain perfect agreement.

A further illustration of the GHD concerns non-equilibrium particle transport. The current statistics has been extensively studied (in the presence of further boundary driving terms) using the large deviation formalism [33, 35, 37]. Here, we look at the domain-wall initial condition:

$$|\psi_{t=0}\rangle = a_1^{\dagger} \dots a_{L/2}^{\dagger} |\Omega\rangle . \qquad (22)$$

The particle transport in a sample QT is shown in Fig. 4.

The GHD description of this initial condition is $n(x, k, t = 0) = \theta(-x)$, where $x = j - L/2$. By linearity of (18), $n(x, k, t) = \int_{x,\infty} G(x', k, t) dx'$ where $G(x, k, t)$ is the solution of (18) with initial condition $G(x, k, t = 0) = \delta(x)$. We can solve this equation in real space/time by randomly sampling the trajectories of classical quasiparticles described above. Namely, one considers a single particle described by position $X(t)$ and momentum $K(t)$, such that:

1. $X'(t) = v(K(t))$ and $X(0) = 0$.

2. $K(t)$ is constant in intervals $[t_0, t_1), [t_1, t_2), \dots$, where $t_0 = 0, t_1 - t_0, t_2 - t_1, \dots$ are independent, and have exponential distribution with average $\overline{t_{i+1} - t_i} = 1/\gamma$.

3. The momenta $K(t) = k_i$ when $[t_i, t_{i+1}]$ are uniformly distributed in $[-\pi, \pi)$ and independent.

We remark that the random trajectory $X(t)$ is exactly that of the non-interacting quasiparticles, as depicted in Fig. 1 (ignoring the dashed lines which represent their entanglement partners, see Section 5.2). It follows from (18) that $G(x, k, t)$ is given by the distribution of the particle

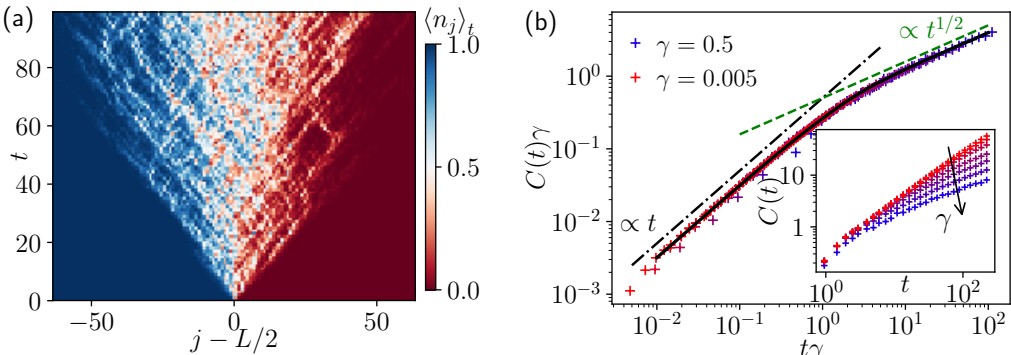

Figure 3: Non-equilibrium transport from a domain wall initial condition. (a) Evolution of local occupation numbers in a typical QT. (b) The averaged current $C(t) = \sum_{j>L/2} \langle n_j \rangle_t$ in a system of $L = 512$. *Main*: Cross-over from ballistic transport $C(t) \propto t$ (the coefficient is set by the $\gamma = 0$ limit) to a diffusive one $C(t) \propto t^{1/2}\gamma^{-1/2}$. The solid curve is the hydrodynamic prediction (23), calculated by sampling 1000 independent random trajectories. *Inset*: Raw data.

position and momentum $x = X(t)$, $k = K(t)$ over the random trajectories. Now, the integrated current across the center $C(t) := \sum_{j>L/2} \langle n_j \rangle_t$ is predicted by GHD as follows:

$$C(t) = \int_0^\infty \int_{-\pi}^\pi \frac{\mathrm{d}x\mathrm{d}k}{2\pi} n(x,k,t) = \int_0^\infty \int_{-\pi}^\pi \frac{\mathrm{d}x\mathrm{d}k}{2\pi} x G(x,k,t) = \overline{\max(0, X(t))}, \qquad (23)$$

where the last expression is an average over all random trajectories $X(t), K(t)$ described above, and can be readily estimated by random sampling. As we show in Fig. 3-(b), the above hydrodynamic prediction compares nicely with numerical simulations. We observe that the transport is ballistic at short times and diffusive at long times, with a crossover at $t \sim 1/\gamma$, which is the mean free time of the quasiparticles.

## 5    Entanglement entropy

In this section we will combine the GHD equation developed above with the quasiparticle pair approach to EE in isolated (integrable) systems. We shall first review the latter approach. Then we present a quantitative theory for the EE in the monitored model, and test it numerically. The results below are the main novel contributions of this work.

### 5.1    Entanglement by quasiparticle pairs: brief review

The quasiparticle pair approach to entanglement entropy emerged originally in applying conformal field theories to isolated quantum systems undergoing quantum quenches [42, 71]. Thereafter, it has been proved to be exact in non-interacting models [65] and more recently in Bethe-Ansatz integrable systems [50, 51]. The basic idea is that, a weakly-entangled but highly excited initial state [e.g., the Néel state (10)] behaves as a reservoir of quasiparticles which are entangled in pairs travelling at opposite momenta $\pm k$, and velocity $\pm v(k)$. These quasiparticles are to be identified with those of GHD in Section 4. During unitary evolution, these pairs propagate coherently and remain entangled, while no entanglement is generated between different pairs. This simplified picture is particularly useful when investigating the evolution in time of EE $S = S_{[x_1,x_2]}$ of an interval $I = [x_1,x_2]$: it is a sum of contributions by

the quasiparticle pairs such that one partner is inside the interval and the other is outside:

$$S(t) = \int_{-\pi}^{\pi} \frac{dk}{2\pi} \int_{Q_{k,t}} dx \, s(x - v(k)t, k, 0), \tag{24}$$

$$Q_{k,t} := \{x \in I : x - 2v(k)t \notin I\}, \tag{25}$$

where $s(x, k, t)$ is the EE contribution from a pair at positions $x$ and $x - 2v(k)$ with momenta $k$ and $-k$, respectively. Note that the velocity $v(k)$ is the same as in (18), and given by (19). For example, in the case of homogeneous ($x$-independent) initial condition, we have the simple formula [51, 52]:

$$S(t) = \int_{-\pi}^{\pi} \frac{dk}{2\pi} \min(2|v(k)|t, \ell) s(k), \text{ where } \ell = x_2 - x_1, \, s(k) = s(0, k, 0). \tag{26}$$

Now, the EE contribution $s(x, k, t)$ is given by the Wigner function Eq. (17) of the initial state via the Yang-Yang entropy [44, 51, 52]:

$$s(x, k, t) = \mathbf{s}[n(x, k, t)], \tag{27}$$

$$\mathbf{s}[n] := -\left[ n \log_2 n + (1 - n) \log_2 (1 - n) \right]. \tag{28}$$

For instance, for the Néel initial condition, $s(k) \equiv 1$. Then, by (26) and (19), $S(t) = (4/\pi)t$ grows linear initially (as $t < \ell/2$) and saturates to $S(t) \to \ell$ as $t \to \infty$. Note that the predictions (24) through (27) are expected to be valid when the interval length is large compared to the lattice spacing but small compared to the total system size, $1 \ll \ell \ll L$. This will also be the expected validity regime of the theory we propose below.

## 5.2 Collapsed quasiparticle pair Ansatz

We now put forward a collapsed quasiparticle pair Ansatz for entanglement growth in the presence of continuous measurement. As in Section 5.1, we assume that the initial condition is highly excited and disentangled (extension to other cases appears nontrivial, see Section 6 for further discussion). Then the Ansatz postulates that:

1. Each quasiparticle pair is destroyed ("collapsed") with probability $2\gamma dt$.

2. When this happens, one quasiparticle of the pair is chosen (with equal probability $1/2$), and a new quasiparticle pair is created at its position, with random momenta $\pm k$, where $k$ is chosen uniformly in $[-\pi, \pi)$.

3. The EE contribution of the new quasiparticle pair created at $(x, t)$ is

$$\tilde{s}(x, t) = \mathbf{s}[n(x, t)], \, n(x, t) := \int_{-\pi}^{\pi} \frac{dk}{2\pi} n(x, k, t), \tag{29}$$

   independently of $k$. Note that eq. (27) still applies to pairs that emanate from $t = 0$.

Note that the pair collapse and creation processes go on indefinitely while different pairs do not interact with each other, as illustrated in Fig. 1. Finally, all the entanglement is produced by such quasiparticle pairs across a bipartite cut of the system. With the above postulates, we can generalize (24) and write down the following quantitative prediction for the averaged EE of an interval $I$:

$$\overline{S(t)} = \tag{30}$$

$$\int_0^t \gamma e^{-\gamma \tau} d\tau \int_{-\pi}^{\pi} \frac{dk}{2\pi} \int_{Q_{k,\tau}} dx \tilde{s}(x - v(k)\tau, t - \tau) + e^{-\gamma t} \int_{-\pi}^{\pi} \frac{dk}{2\pi} \int_{Q_{k,t}} dx \, s(x - v(k)t, k, 0),$$

where $Q_{k,t}$ is defined in (25). In the RHS, the first term accounts for measurement-created pairs and the second term accounts for those of the initial condition that are not collapsed yet.

The collapsed quasiparticle pair Ansatz is motivated by consistency with the quasiparticle pair approach in isolated systems and the GHD under continuous measurement of Section 4. Indeed, the quasiparticles in GHD are identified with the chosen ones in the pair picture, as is clear from Fig. 1. Physically, the Ansatz tries to capture the effect of continuous measurement on the conditional state, by approximating it with a projective measurement applied with rate $\gamma$. Indeed, the latter projectively collapses the quasiparticle wavefunction; the collapsed wavefunction is localized at a single lattice site, and thus emits an entangled quasiparticle pair, with maximal momentum uncertainty. This is exactly the dynamical rules described above. We expect that the Ansatz applies specifically to the entanglement of the conditioned state, and to the specific unravelling (3) (see Section 6 for discussion on other unravellings).

It should be also clear that the above physical arguments are heuristic and not a rigorous justification of the Ansatz, which is a nontrivial question left to future investigation. In what follows, we assess its correctness by several numerical tests.

### 5.3 Numerical tests

In the numerical tests below, we compute the EE using (16) from QT simulations (by the method of Section 3), average the result over many QT realizations, and compare the outcome with the prediction obtained from the above Ansatz.

#### 5.3.1 Stationary regime

In the long time limit, the Wigner function becomes a constant and equal to the filling factor:

$$n(x, k, t \to \infty) \longrightarrow n = N/L, \tag{31}$$

for any $\gamma > 0$. Then Eq. (30) simplifies to the following scaling form for an interval of length $\ell$:

$$\overline{S}\Big|_{t \to \infty} = \ell f(\ell \gamma), \text{ where } f(x) = \mathbf{s}[n] \int_{-\pi}^{\pi} \frac{dk}{2\pi} \frac{2|v(k)|}{x} \left[1 - e^{-x/(2|v(k)|)}\right]. \tag{32}$$

As we show in Fig. 4, the above scaling relation describes exactly the stationary EE in numerical simulations for a wide range of measurement rates, so long as $1 \ll \ell \ll L$, as expected. For any fixed $\ell$, as $\gamma \to 0$, $x \to 0$ and the EE density $f(x) \to \mathbf{s}[n]$ becomes the thermal entropy, which is expected in a system without measurement. However, for any $\gamma > 0$, as $\ell \to \infty$, we have

$$f(x) \overset{x \gg 1}{\simeq} \frac{c}{x} \text{ where } c := \mathbf{s}[n] \int_{-\pi}^{\pi} \frac{dk}{2\pi} 2|v(k)| \tag{33}$$

is a constant. We conclude that the EE follows an area law in the presence of any measurement:

$$\overline{S}\Big|_{t \to \infty, \ell \to \infty} = \frac{c}{\gamma}. \tag{34}$$

Intuitively, the reason is that quasiparticle pairs have finite life time $\sim 1/\gamma$ in average, so entanglement cannot be built beyond the scale $\sim \max_k |v(k)|/\gamma$.

#### 5.3.2 Homogeneous quench

We now consider the time evolution of EE, starting from a product initial state with uniform density $n \in (0, 1)$ on the macroscopic scale. For example, the Néel state (10) is an example

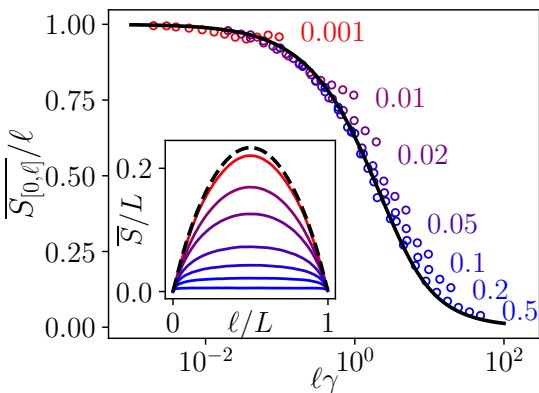

Figure 4: Averaged EE in the stationary regime on $[0, \ell]$ in a system of size $L = 512$ and filling factor $n = \frac{1}{2}$, with varying measurement rate $\gamma = 0.001, 0.01, 0.02, 0.05, 0.1, 0.2, 0.5$ (from red to blue, indicated besides each data set). The main plot collapses the numerical data (in circles) according to the scaling relation (32), which is satisfied as long as $1 \ll \ell \ll L$. The prediction (32) is plotted in thick curve. The inset shows the raw data (plotted in solid curves with the same color codes), compared to the $\gamma = 0$ case drawn with thick dashed curve.

with $n = \frac{1}{2}$, and the state

$$|\psi_0\rangle = \left[\prod_j a_{3j}^\dagger\right] |\Omega\rangle \tag{35}$$

has $n = \frac{1}{3}$. In either case, the Wigner function $n(x, k, t) \equiv n$ is a constant and has no evolution at a macroscopic level, while the EE grows from 0 at $t = 0$ to the stationary value (32). Now, the precise time dependence can be predicted by the Ansatz (30):

$$\overline{S(t)} = \int_{-\pi}^{\pi} \frac{dk}{2\pi} \frac{2|v(k)|}{\gamma} \left(1 - e^{-\gamma \min(t, t_*)}\right) \mathbf{s}[n], \quad \text{where } t_* = \frac{\ell}{2|v(k)|}. \tag{36}$$

Fig. 5 shows its agreement with the numerics. Observe that in a sufficiently large interval $\ell > 2 \max_k |v(k)| t$, (36) further simplifies to a scaling form (as shown in Fig. 5):

$$\overline{S(t)}\gamma = c\left(1 - e^{-y}\right), \quad y = \gamma t, \tag{37}$$

where $c$ is defined in (33). The time scale $1/\gamma$ is that of EE suppression by measurement: The EE grows as in an isolated system when $t \ll \gamma$ and tends to the stationary area-law value $c/\gamma$ (34) as $t \gg \gamma$.

Another consequence of the Ansatz is that the EE fluctuations among individual QT's is rather featureless. Since the EE is a sum over quasiparticles pairs, which have independent dynamics, the EE statistics is simply Gaussian, which we observed numerically. So, although the QT dynamics (3) is random and non-linear, it does not resemble the one of random unitary circuits [72] which generate nontrivial EE fluctuations and are supposed to describe noisy dynamics of generic systems [62].

### 5.3.3 Two-reservoir quench

Finally we consider initial conditions that are junctions of two homogeneous ones (as in Section 5.3.2), with different densities: in GHD,

$$n(x, k, 0) = \begin{cases} n_L & x < 0 \\ n_R & x > 0 \end{cases}. \tag{38}$$

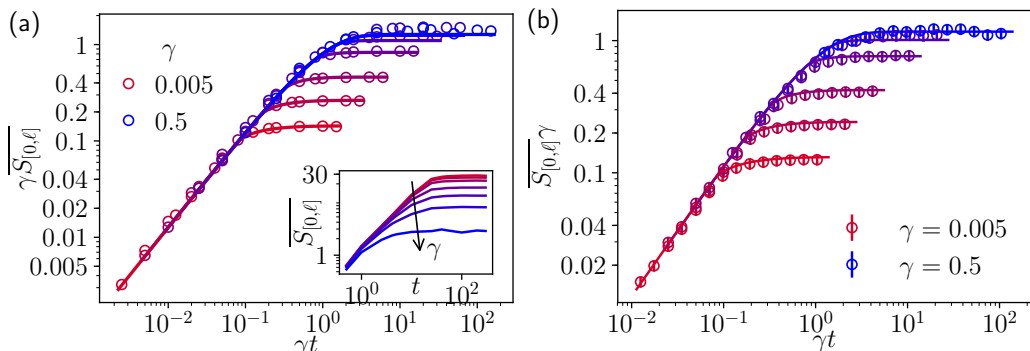

Figure 5: Growth of EE of an interval $[0,\ell], \ell = 32$ in a chain of size $L = 512$, with measurement rates $\gamma = 0.005, 0.01, 0.02, 0.05, 0.1, 0.5$ (from red to blue). The data points are numerical results, the curves are the prediction (36). The main plots highlight the short-time scaling form (37). The inset plots the same data in naïve units. The initial states are (a) Néel state (10) with $n = \frac{1}{2}$ and (b) the state (35) with $n = \frac{1}{3}$.

By (18), $n(x, k, t)$ depends on all three variables $x, k$ and $t$, therefore the present setting is a more stringent test of the Ansatz, in particular, of Eq. (29). On the other hand, the analytical prediction (30) can no longer be significantly simplified, and needs to be integrated numerically [4]. In Fig. 6-(a), the obtained prediction compares nicely with data from direct QT simulation. We observe that short time EE growth still satisfies the scaling form $\overline{S(t)}\gamma = f(\gamma t)$ (compare to (37)). This is not hard to show from the Ansatz, by observing the invariance of GHD (18) under scaling the scaling $\gamma \rightsquigarrow a\gamma$, $t \rightsquigarrow t/a$, $x \rightsquigarrow x/a$.

Finally we push beyond the expected validity of the Ansatz and revisit the domain-wall case (22), corresponding to $n_L = 1$ and $n_R = 0$. Note that this is qualitatively different from the cases where $0 < n_{L,R} < 1$: entanglement is only produced from the junction, as the system only evolves significantly inside the light cone $|x| < \max_k |v(k)|t$, see Fig. 3-(a). In the free-fermion ($\gamma = 0$) case, the EE has a peculiar logarithmic growth in time [73], which is not captured by the quasiparticle pair approach reviewed in Section 5.1. From QT simulation, we observe that continuous measurement significantly enhances the EE growth in this case as $t \gtrsim 1/\gamma$, as shown in Fig. 6-(b). Interestingly, even in this intricate case, the collapsed quasiparticle pair Ansatz describes reasonably well the EE evolution, and becomes increasingly accurate as the measurement becomes more dominant. This is likely because the conformal invariance [73] of the free-fermion case is broken by the measurement, and measurement-created quasiparticle pairs dominate the EE production at later times.

From the above tests, we may conclude that the collapsed quasiparticle pair Ansatz is at least an accurate approximate theory of EE dynamics of monitored free fermions. We note that, in general, entanglement entropy in many body systems is nontrivial to calculate analytically, even when the system itself is exactly solvable or even non-interacting. In this respect, the performance of the Ansatz is remarkable given the conceptual simplicity underlying it. We also remark that, the continuous measurement is modelled in the Ansatz as discrete but projective measurements applied to a set of space-time points generated by a Poisson point process with density function $\gamma dx dt$. While one may expect the qualitative similarity between continuous and dilute projective measurements, the quantitative equivalence suggested by our results is much less obvious.

---

[4]In practice this is done in two steps: we first solve the GHD equation (18) by the method in Section 4, and then calculate the triple integral (30) by Monte Carlo sampling.

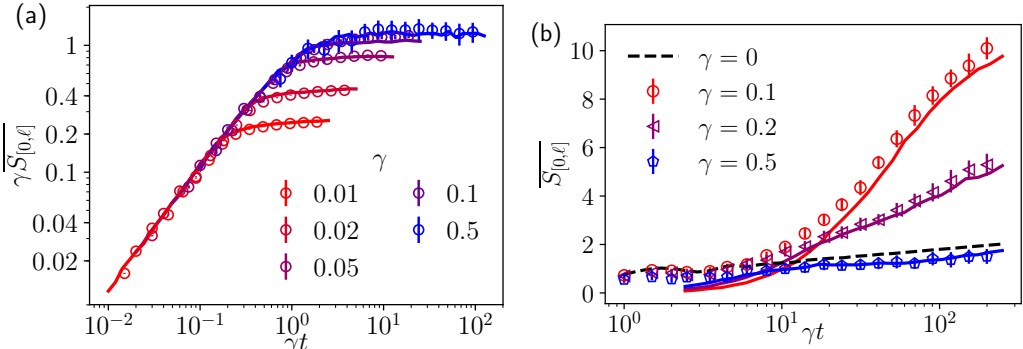

Figure 6: The evolution of EE in an interval $[0,\ell]$ ($\ell = 32$ in an open chain of $L = 512$) from junction initial states. QT simulation results are plotted with markers and predictions from the Ansatz in solid curves. The colors indicate the value of $\gamma$. (a) $n_L = \frac{2}{3}$, $n_R = \frac{2}{3}$. The data are plotted to show the short-time scaling relation (37). (b) $n_L = 1$, $n_R = 0$ (domain wall initial condition). The dashed curve shows the EE without measurement in the half-infinite interval $[0, +\infty)$ which grows as $\propto \ln t$.

## 6 Discussion

The main result of this work is a quantitative description of the entanglement dynamics in a free fermion chain under continuous measurement of all occupation numbers, called the collapsed quasiparticle pair Ansatz. Its principle consequence is that the volume law entanglement in short-range free fermion systems is unstable under the perturbation of arbitrary weak measurement if it is applied everywhere. The reason for this is that entanglement entropy is carried by quasiparticle pairs, which have a finite coherent life time under measurement. Therefore, long range entanglement is suppressed, as long as the quasiparticle velocities are bounded. Although our numerical tests are performed on a 1d chain, we believe that the Ansatz also applies to higher dimensions, leading to the same physical conclusion. The Ansatz is expected to be valid for highly excited and disentangled initial conditions. We have seen that it does not describe exactly the EE evolution from the domain wall quench. We observed also that the Ansatz prediction is not quantitatively correct if the initial state is a high energy eigenstate. The destruction of its volume law entanglement by measurement is probably not described by pair collapsing. After all, it should not be a surprise that the Ansatz we proposed is not the complete theory, since even free fermion models without measurement give rise to entanglement dynamics beyond the quasi-particle pair picture [74, 75].

The quantitative study of this work focused on the bipartite entanglement entropy. However, this quantity involves both many-body quantum correlations and few-body ones, and leaves us rather ignorant of the structure of the entanglement. To take a first qualitative look into that, let us consider the quantum correlations between pairs of lattice sites [76]. This can be quantified by the mutual information,

$$I(x_1, x_2) = S_{\{x_1\}} + S_{\{x_2\}} - S_{\{x_1, x_2\}}, \quad x_1 \neq x_2, \tag{39}$$

where $S_A$ is defined as in (24). In general, $0 \leq I(x_1, x_2) \leq 2$ and the maximal value is obtained if and only if the qubits at $x_1$ and $x_2$ form a maximally entanglement pair and are otherwise decoupled from the rest of the system. We measured the mutual information between all pairs for a few typical QT's in the stationary regime. Some correlation snapshots obtained in this way are shown in Fig. 7. We observe that, compared to the free fermion case, introducing continuous measurement significantly enhances the correlation between a few random pairs of site separated by a short distance. (The characteristic length beyond decreases as $\gamma$ increases.)

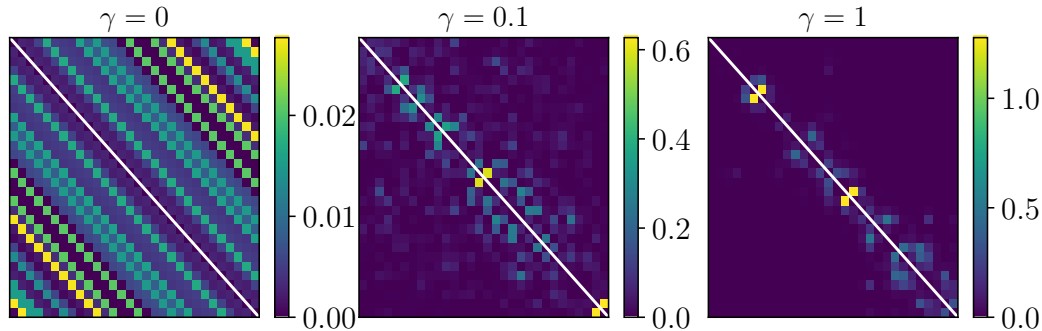

Figure 7: Color plot of the matrix of mutual information $[I(x_1, x_2)]_{x_1,x_2=1}^{30}$ (in a periodic chain of size $L = 128$) for a few typical realizations of QT's, obtained at $t = 100$ after a quench from the Néel state. On the diagonal $x_1 = x_2$, marked by a white line, the plot is meaningless. The square lattice grid is a guide to the eye. QT's from different values of the measurement strengths $\gamma$ are plotted in the three panels. Note that they have different color bar scales, to accommodate the pronounced increase of the magnitude of the maximal mutual information as continuous monitoring is introduced.

This indicates that, although the total amount of entanglement is reduced by measurements (as testified by the area-law bipartite EE), the entanglement in continuously monitored QT's are more distilled, and potentially more usable as a quantum resource. This raises the fascinating question of tow to access the entanglement produced in QT's.

Our approach to entanglement entropy depend crucially on the particular unravelling (3) of the Lindblad equation (6). In contrast, the GHD applies to the mean state, and therefore to the average of linear functionals of the conditional states (QT's) under arbitrary unravelling [see (7)]. For instance, the *unitary unravelling* corresponds to the following SSE:

$$\mathbf{i}\,\mathrm{d}|\psi_t\rangle = H\mathrm{d}t|\psi_t\rangle + \sum_{i=1}^{L}\left(\sqrt{\gamma}n_i\mathrm{d}W_t^i - \frac{\mathbf{i}\gamma}{2}n_i^2\mathrm{d}t\right)|\psi_t\rangle. \tag{40}$$

It implies the same Lindblad equation (6) for the mean state $\overline{\rho_t}$, so the particle transport is still governed by the same GHD equation (18). However, the entanglement dynamics of the QT's is qualitatively different. Indeed, while the continuous measurement term in the SSE (3) tends to disentangle the site $i$ from the rest of the system, the dephasing term in (40) (second term of RHS) amounts to applying one-site random unitary gates, which do not change the entanglement themselves, but they do affect the entanglement when combined with free fermion evolution. This makes it more difficult to propose an Ansatz describing the QT entanglement. Nevertheless, using the numerical method of Section 3 (properly modified), we found that, starting from low-entanglement state, the EE grows as $\sim \sqrt{t}$ up to a volume law, see Fig. 8 for a numerical result. This behavior is close to the random fermion model studied in Ref. [72]. Therefore, the entanglement dynamics can change considerably from one unravelling scheme to another. The dependence of EE on the unravelling scheme has been studied in few body models [31], the extension to of many-body systems is a nontrivial task left to future work.

Going beyond non-interacting models, a question of great interest is the effect of interactions in stabilizing the volume law entanglement in continuous monitored systems. In fact, stability of volume law entanglement may be a diagnosis of quantum thermalization and chaos; finding such a diagnosis for general quantum many-body systems is a nontrivial task, and is one goal of much recent research on many-body quantum chaos, see e.g. [77–85].

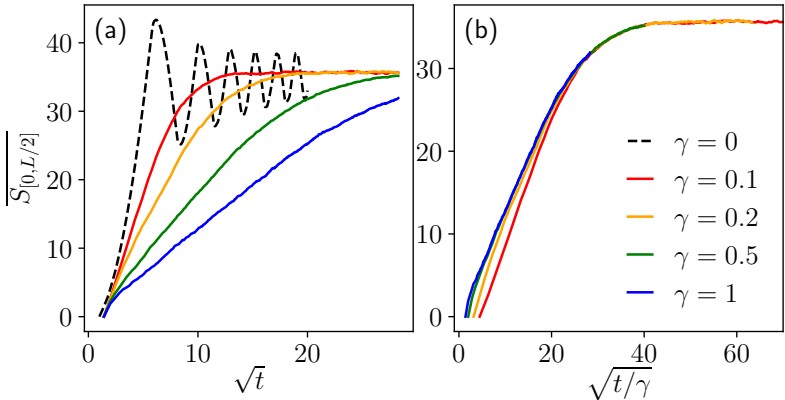

Figure 8: Averaged EE in the interval $[0, L/2]$ (in an periodic chain with $L = 128$) of quantum trajectories satisfying the stochastic Schrödinger equation (40) for various values of $\gamma$. The initial condition is the Néel state (10). About 20 QT's are averaged over for each curve. The simulation is performed with a method similar to that described in Section 3. (a) The averaged EE grows as $\propto \sqrt{t}$, until reaching a volume-law value nearly identical to the free-fermion saturation EE ($\gamma = 0$, dashed curve). (b) The same data is plotted against a re-scaled time $\sqrt{t/\gamma}$. We observe a collapse which improves as $\gamma$ increases. A quantitative understanding of such a behavior is an open question.

So far, the most studied case is that of strong and non-integrable interactions [27–30]. There, entanglement is genuinely many-body (instead of being produced by quasiparticle pairs), making possible a volume law phase and a entanglement transition by tuning $\gamma$: their existence is plausible but not completely proved. We believe it will be instructive to study a two-dimensional phase diagram by introducing the interaction strength ($g$) as additional parameter: what is the minimal $g$ required to stabilize the volume law phase (or, do the free fermion results extend to weakly interacting fermions)?

Now, when the interaction is integrable (by Bethe Ansatz), the situation is completely different. Indeed, the quasiparticle pair picture reviewed in Section 5.1 applies to interacting integrable systems [50–52], with the additional complexity that, quasiparticles can have several species and their velocity depends on the local quasiparticle density (the details depend on the thermal Bethe Ansatz of particular models [45–48]). Therefore, it is in principle possible to apply the method in this work to these systems. As a first step of this, we will need to extend the GHD equation (18), in particular, its measurement term. Doing this from first principle is harder than deriving (18), and some approximations and/or modification of the form of monitored observables would be necessary for a tractable GHD equation to be valid. Despite this technical caveat, on physical grounds, we expect the main effect of the measurement should still be the scattering of quasiparticles, as depicted in Fig. 1. Then, the collapsed quasiparticle pair Ansatz could be still qualitatively correct, *mutatis mutandis*, and leads to a destruction of volume-law entanglement. We defer a detailed analysis of these issues to a future study.

# Acknowledgements

We thank Denis Bernard, Federico Carollo, Adam Nahum, and Michael Knap, for insightful discussions and useful suggestions on the manuscript. We thank the anonymous referees for instructive comments.

**Funding information** The authors acknowledge support from the Alexander von Humboldt foundation and the Agence Nationale de la Recherche contract ANR-14-CE25-0003-01 (AT), ERC synergy Grant UQUAM and DOE grant DE-SC0019380 (XC), and EPSRC Quantum Matter in and out of Equilibrium Ref. EP/N01930X/1 (ADL). Numerical computations have been performed thanks to facilities at Laboratoire de Physique Théorique et Modèles Statistiques, Orsay, France.

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
