# Peer review of "Entanglement in a fermion chain under continuous monitoring"

_SciPost Physics, doi:SciPost Phys. 7, 024 (2019)_

## Round 4 · Referee Report · Anonymous (Referee 2) · 2019-7-1

Strengths

1) Clear and well written 2) Interesting results on a timely topic 3) The prediction is based on a simple picture and this is clearly presented

Weaknesses

1) It is not clear how general this can be made. This is however also honestly discussed by the Authors.

Report

The manuscript deals with the problem of understanding entanglement behaviour in single dynamical trajectories of open quantum system. In particular, the Authors focus on a quantum system consisting of a free-fermion chain under continuous monitoring of local occupation numbers. The time-evolution of the state of such a system, while being described on average by Lindblad master equations, is stochastic and the state itself at any time t is a random observable.

In order to study this problem the Authors derive a Generalised Hydrodynamic Description (GHD) by which they can first predict the dynamical behaviour of linear observables of the state (i.e. observables that are captured by the Lindblad dynamics of the quantum system). This prediction is confirmed by numerical simulations.

The understanding at the basis of the GHD combined with the quasi-particle pair approach in integrable systems further allows the Authors to formulate a description of the dynamics of the entanglement entropy in stochastic trajectories by means of a simple collapsed quasi-particle Ansatz. This postulates that a particle of a quasiparticle pair is destroyed by effect of the measurement (with a suitable probability) and is substituted by a new quasiparticle pair with random momentum that begins to propagate from the position where the previous particle was destroyed.

From these rather simple rules the Authors derive a prediction for the average (over quantum trajectories) entanglement entropy dynamics. As the latter is not a linear function of the state, there is no hope in extracting this information from the Lindblad time-evolution.

The prediction is checked against extensive large scale numerical results that are possible in this case as the unraveled dynamics of such an open quantum system is “non-interacting”.

The agreement between the numerics and the theoretical prediction is remarkable. A relevant observation is that the entanglement entropy obeys for long times an area law, for any finite amount of the measurement rate \gamma.

I believe that the manuscript is well written and every conceptual step is introduced in a very clear way by the Authors. The problem addressed here is timely and, in my opinion, of extreme interest to a wide audience of physicists, e.g. working on open quantum systems or stochastic processes, but also on GHDs of integrable system dynamics.

I think that the results presented in the manuscript represents an important step forward in understanding the dynamics of truly quantum features in stochastic quantum system as those arising from unraveled dynamics of Lindblad quantum systems.
For these reasons I strongly recommend publication.

I have, however, few minor comments and a slightly major one. I would suggest the Authors to address these.

Minor:
— 1) Eq. (6) a dot instead of a comma at the end of the equation;

— 2) I guess there should be a normalisation factor in eq.(12);

— 3) Just after Eq. (14) the Authors write “ For this, we assume the particle number is conserved in the Gaussian state: this is done by considering particle number conserving Gaussian initial states, …”
I think here the Authors mean that the initial state has a fixed number of particle, but I don’t think this is clearly expressed by saying that the initial state conserves the number of particle.

— 4) Eq. (22) there should be a minus in the term between equal sings. Otherwise that looks proportional to a cosine.

Slightly major:

— 5) I seem to have a problem with Eqs. (8-9-10) and with the paragraph discussing these.
First, the authors introduce O[\rho_t]=\rho_t^{\otimes 2}. The notation is not explicitly defined, so I guess this means

\rho_t^{\otimes 2}=\rho_t \otimes \rho_t.

If this is the case the trace of this object (which since it is not otherwise defined I take it as the usual trace over the double space) is

\Tr(\rho_t \otimes \rho_t)=\Tr(\rho_t) \Tr(\rho_t) = 1 given that one wants \Tr(\rho_t)=1.

As such I think that this cannot be used as a figure of merit for purity of the global state. The usual purity is defined as \Tr(\rho_t^2). In any case, since the state in trajectories is always a pure state —if the initial state is— this is also always 1. The average of the purity over stochastic trajectories of pure states is 1. This is just a matter of definitions.

What confuses me is the following. Taking the operator \rho_t \otimes \rho_t, the Authors derive the Lindblad in Eqs. (8), by exploiting Ito rules and the average over quantum trajectories (QT). I am fine with the pieces of Eq. (9) and I understand that (10) must come from the Wiener cross-interactions between the two parts of the tensor product. The stochastic evolution of \rho_t can be written as (with appropriate identifications of the \mathcal{K}_i[\rho_t])

d\rho_t=\mathcal{L}[\rho_t] dt +\mathcal{K}_i[\rho_t] dW^i_t .

Then the term (10) must come from

\bar{ \mathcal{K}_i[\rho_t] \otimes \mathcal{K}_i[\rho_t] }

where \bar is as in the manuscript the average over quantum trajectories.

In particular, in computing this, one must take expectation over trajectories of three-point functions of the stochastic state \rho_t since one has terms like

\bar{ n_j \rho_t \otimes \rho_t \Tr(\rho_t n_j) };

it seems to me that the only way to get the Eq. (8) with the term (10) that the Authors provide is by factorizing the above expectation into

n_j \bar{ \rho_t \otimes \rho_t } \bar{ \Tr(\rho_t n_j) }.

Is there a clear reason why this can be done? Otherwise one would expect the term (10) to be much more complicated.
I also notice that the generator L^2 is time-dependent as it depends on \bar{<n_j>}_t; this is not mentioned when discussing Eq.(8) but this should be highlighted as it is certainly not expected and not transparent from Eq.(8).

I would encourage the Authors to expand the discussion about the derivation of Eq. (8) as I think that, as it is, this might generate quite a confusion.
If this discussion becomes too involved maybe the Authors could remove this —as this is not really needed to follow the paper— and cite other results, e.g. [Phys. Rev. B 98, 195125 (2018); Phys. Rev. B 98, 184416 (2018)], when discussing that the dynamics of \bar{ \rho_t \otimes \rho_t } is not given by the Lindblad dynamics which evolves linear function of the state, at least not in general.

Requested changes

Suggested changes are discussed in the Report.

---

## Round 4 · Referee Report · Anonymous (Referee 1) · 2019-7-1

Strengths

1) Studies subject of recent interest 2) Gives a clear physical picture behind results 3) Verifies results by exact numerical calculations 4) Proposes theoretical ansatz for a simplified description, but nevertheless comments on situations when it does not work.

Weaknesses

No real weaknesses, perhaps add some additional context along the lines suggested.

Report

In the manuscript "Entanglement in a free fermion chain under continuous monitoring" the authors study pure-state bipartite entanglement of quantum trajectories representing evolution of free fermions (tight-binding model) under the influence of independent density measurements on all sites.

They find that the entanglement between a section of length l and the rest of the chain in the scaling limit l-> infty and at a fixed dissipation strength gamma never displays a volume law (as would be the case under unitary evolution alone). Finite dissipation destroys volume law entanglement. Nevertheless, on a finite lengths l one has nonzero entanglement scaling as 1/gamma and being essentially given by correlations within a coherence length. They demonstrate these results by exact numerical simulation of initial Gaussian states and by a heuristic ansatz employing generalized hydrodynamics picture of propagating (quasi)particle pairs interspersed by a collapse due to measurements.

The questions related to the ones addressed have been considered in a number of recent papers and are of considerable interest. The results seem new and the ansatz proposed could turn out to be useful more generally. The paper is nicely written and easy to read. I would therefore recommend its publication.

I have only a couple of minor comments and questions for the authors to consider. Mostly I would suggest to try to add some additional explanations in order to put the work into wider context and explain why the subject is studied.

Requested changes

Specific questions in order of appearance:

1) Perhaps already in the introduction mention a bit more why you study quantum trajectories. Nonspecialist might not be familiar with the difference of Lindblad vs. QT and so e.g., explicitly mentioning (like you do later) that for the question studied one would give the entanglement of the average state, while QT gives the average entanglement would be instructive.

2) A bit connected to 1). What decides which of the two settings (Lindblad, QT) is the "correct" one? One can imagine that in an experimental setting, if one would do a tomography of a state, and if measurements take same time, one would get some time-averaged state which might then be closer to what one would get out of Lindblad equation?

3) In Sec.2.2, the footnote 2 is very important. Rather often one sees that people blindly use Von Neumann entropy of a mixed state and at the same time speak about entanglement. I would actually be even more explicit in the last sentence "..is not a good measure..". vNE is not a measure of entanglement at all! It measures mixedness. (perhaps remove the word "good")

4) Comment 3) begs another question. Much of interest in entanglement in last decades stems from its quantumness and possible uses in quantum information as a resource. However, one must be aware that physically relevant are only few-body observables -- one can not measure many-particle correlations. Similar is situation with entanglement: bipartite entanglement between thermodynamically large pieces is physically (experimentally) not relevant. Of course, it can be a useful concept on paper for some theoretical studies, like thermalization. One can show that even if one has a lot of such bipartite entanglement it is not a useful resource (e.g., arXiv:0812.3001, arXiv:0810.4331). This brings us to the question what about few-particle entanglement? For instance, the simplest would be a 2-body entanglement between say site j and k. Such a quantity might be first of experimental relevance, and second it probably has some (interesting?) dependence on the distance |j-k| [in a driven Lindblad setting and the model studied it was briefly considered in arXiv:1112.4415].

5) On p.7 lambda is set to 1/2. Is there some interesting dependence on lambda? It seems that on the level of QT's one does not have just a single parameter lambda/gamma.

6) Sec.6.: observation that different unravellings can give different entanglement dynamics is interesting. I would like to see/hear more about it. Unless authors plan to study this further it would be interesting to show e.g. a figure with the data they mention.

---

## Round 5 · Referee Report · Anonymous (Referee 3) · 2019-8-2

Report

I am satisfied with the changes made by the Authors and I recommend publication of the manuscript in SciPost.

---

## Round 5 · Referee Report · Anonymous (Referee 4) · 2019-8-10

Report

Authors have addressed my questions/suggestions and I recommend publication of the manuscript.

---

## Round 5 · Author Response

Dear Editor and Referees,

We are thankful for your interest in the manuscript and helpful comments.

In the submitted revision, we strived to address all your suggestions. In particular, we added two new figures. Please see the list of changes for details.

We hope that the revised manuscript meets the criteria of SciPost.

Yours sincerely,

---

## Round 5 · List of Changes

major changes:
- added a figure on the entanglement growth with unitary unravelling to support the discussion (point 6 of Report 1).
- added a figure on the two-point mutual information of some representative quantum trajectories and discussed the meaning of the result (point 4 of Report 1).
- removed the previous flawed calculation in section 2 on the evolution of tensor squared.
Used instead a simpler argument (inspired by the referee's suggestion) to make our point (point 5 of Report 2).
- added a paragraph in introduction discussing significance of our study from the perspective of simulation of open quantum systems (point 1-2 of Report 1).

minor:
- fixed points 1 - 4 of Report 2.
- removed "good" from the footnote, following point 3 of Report 1.
- adding an explanation why fixing \lambda = 1/2 is ok (changing it and \gamma amounts to rescaling time; point 5 of Report 1).

---

## Editorial Decision

published